# Quantification of One-Year Gypsy Moth Defoliation Extent in Wonju, Korea, Using Landsat Satellite Images

**Won-IL Choi \*, Eun-Sook Kim, Soon-Jin Yun, Jong-Hwan Lim and Ye-Eun Kim**

Division of Forest Ecology, National Institute of Forest Science, 57 Hoegi-ro Dongdae mun-gu, Seoul 130-712, Korea; drummer12@korea.kr (E.-S.K.); yunsj86@korea.kr (S.-J.Y.); limjh@korea.kr (J.-H.L.); yaeeun0910@naver.com (Y.-E.K.)
**\*** Correspondence: wchoi71@korea.kr; Tel.: +82-(2)-961-2604; Fax: +82-(2)-961-2629

**Abstract:** We quantified the extent and severity of Asian gypsy moth (*Lymantria dispar*) defoliation in Wonju, Korea, from May to early June in 2020. Landsat images were collected covering Wonju and the surrounding area in June from 2017 to 2020. Forest damage was evaluated based on differences between the Normalized Difference Moisture Index (NDMI) from images acquired in 8 June 2020 and the prior mean NDMI estimated from images in June from 2017 to 2019. The values of NDMI ranged from −1 to 1, where values closer to 1 meant higher canopy cover. The NDMI values for 7825 ha of forests were reduced by more than 0.05 compared to the mean NDMI values for the prior 3 years (2017 to 2019). The NDMI values of 1350 ha of forests were reduced by >0.125 to 0.2, and the NDMI values for another 656 ha were reduced by more than 0.2. A field survey showed that these forests were defoliated by gypsy moth and that forests with NDMI reductions of more than 0.2 were heavily defoliated by gypsy moth. A 311 ha area of Japanese larch (*Larix kaempferi*) was severely damaged by gypsy moth and the proportion of larch damaged was higher than that of other tree species. This intense damage to larch suggests that gypsy moths preferentially attack Japanese larch in Wonju. Our study shows that the use of NDMI values to detect areas defoliated by gypsy moth from satellite images is effective and can be used to measure other characteristics of gypsy moth defoliation events, such as host preferences under field conditions.

**Keywords:** gypsy moth; Landsat; normalized difference moisture index (NDMI); remote sensing; satellite image

## 1. Introduction

The quantification and mapping of forest areas damaged by outbreaks of forest pests offer insights into the causes of outbreaks and their impacts on forest stands, which is information that can be used for pest management [1]. Several methods exist for the quantification and mapping of forest insect damage, including field surveys, aerial sketching, and remote sensing [1,2]. Among these options, remote sensing using satellite images is considered to have merit due to the lower amount of labor required, the speed of data processing, and improved accuracy [1].

The quantification of area damaged by forest pests using remote sensing technologies has been extensively reviewed [3–8]. In Canada, forest areas damaged by eight major forest pests, including mountain pine beetle (*Dendroctonus ponderosae* Hopkins), forest tent caterpillar (*Malacosoma disstria* Hübner), and gypsy moth (*Lymantria dispar* (L.)), were quantified using satellite images from MODIS, Landsat, and Kompsat-3, etc. [3]. That study also showed that the Normalized Difference Moisture Index (NDMI) was useful in detecting damage by mountain pine beetle. The forest area damaged by gypsy moth outbreaks caused by the spring drought from 2015 to 2017 in southern New England, USA, was quantified and mapped using Landsat images and the index of forest canopy greenness [5]. In Pennsylvania and Maryland, USA, Landsat satellite images were used to relate the size

of gypsy moth outbreaks to nitrogen concentrations in stream water in the Fifteenmile Creek watershed [9].

Gypsy moth is native to Europe and Asia [10]. In North America, gypsy moth was introduced from Europe in the 1860s, and is considered a major forest pest and a significant invasive species in North America [5]. In contrast, gypsy moth in Korea is the Asian subspecies and is only an occasional pest [11,12], although local outbreaks were observed in the 1990s and 2000s [13,14]. Until the 1990s, there was little information about the outbreak of gypsy moth in Korea, except for a small outbreak in 1200 deciduous trees in Seoul in 1959 [12]. Gypsy moth in Korea has a wide host range, including both deciduous and coniferous trees, but the host preference of the moth has not been recorded [14]. The fact that the egg masses of gypsy moth were collected in mainly *Quercus* spp. forests in the early 1990s suggest that *Quercus* spp. are potentially preferred hosts in Korea [12]. Similarly, gypsy moth in North America prefers *Quercus*, *Populus*, and *Salix* spp., and outbreaks of the moth usually occurred in these forests [10]. In Korea, gypsy moth outbreaks are more commonly observed in forests near human residences than in natural forests [14].

The aim of this study was to quantify the extent and severity of gypsy moth defoliation in and around Wonju, Korea, in 2020 using Landsat Satellite Images and NDMI, documenting the extensive outbreak of gypsy moth reported there by Jung et al. [15]. By analyzing defoliated areas, we determined the pest's field host preference and the physical characteristics of the areas defoliated by gypsy moth in Korea.

## 2. Materials and Methods

### 2.1. Study Site

The forest area examined was located in Wonju, Gangwon-do, Korea (37°21′5″ N, 127°56′43″ E). Wonju is an urban area with 350,000 residents and an area of 867.30 km$^2$. The climate at the survey area is continental, with an annual mean temperature of 12.3 °C and an annual precipitation of 1276 mm over the last decade (Korean Meteorological Administration, http://www.weather.go.kr (accessed on 26 April 2021)).

### 2.2. Satellite Image Acquisition and Analysis

To assess the area of forest damaged by gypsy moth, serial Landsat images (Path/Row 115/34) covering Wonju and the surrounding area from 2017 to 2020 were obtained from the USGS EarthExplorer homepage (https://earthexplorer.usgs.gov (accessed on 26 April 2021)). Basically, one scene of Landsat covered Wonji city and its vicinity. To estimate baseline information for each year, 20 images per year were collected. Therefore, 60 images were collected to estimate 3 prior years baseline information. The Landsat-8 images used in this study were orthorectified, radiometrically corrected, and included a cloud mask. Landsat images from 2017 to 2019 were used to produce baseline information before the event using a pixel-based time-series gap filling method [16]. Additionally, images from 2020 were used to evaluate the damage caused by the event compared to the past. Damaged forest areas were evaluated based on the difference between the Normalized Difference Moisture Index (NDMI) estimated from images acquired in 8 June 2020, and the mean NDMI estimated from images acquired in June from 2017 to 2019. NDMI is an effective index to evaluate the extent of forest damage from natural or anthropogenic disturbances [3]. The values of NDMI ranged from −1 to 1, with values closer to 1 meaning higher canopy cover [17]. The mean NDMI value from 2017 to 2019 (pre-defoliation) was calculated by the methods proposed by Kim et al. [16], such as NDMI = (NIR−SWIR1)/(NIR + SWIR1), where NIR is the pixel value from the near infrared band (0.85–0.88 μm) and SWIR1 is the pixel value from the short-wave infrared 1 band (1.57–1.65 μm) [18]. A mean NDMI map from 2017 to 2019 was produced by the time-series gap-filling method using clear sky pixels [16]. This map provided baseline information before the event.

Through the visual inspection of satellite images for Wonju in June 2020, forest areas with a red or brown color were selected, and those areas where the NDMI value had decreased by 0.2 compared to the values from 2017 to 2019 were noted. To minimize

errors during the imaging process, areas where the NDMI was reduced by over 0.05 were considered to be defoliated by gypsy moth. The degree of forest defoliation was classified into three levels: (1) Severe (NDMI reduction of $\geq$0.2), (2) moderate (NDMI reduction of >0.125 to 0.2) or (3) light (NDMI reduction of 0.05 to 0.125). The forest types used for the classification of stands were based on a forest type map (http://www.forest.go.kr (accessed on 26 April 2021)). The Korean forest type map was produced with 5-year intervals using aerial photography and field observations at two spatial scales: 1:5000 and 1:25,000 [19]. In our analysis, a 1:5000 scale map was used. Forests with >75% coniferous trees or >75% deciduous trees were classified as coniferous or deciduous. Coniferous stands were further divided into Korean red pine (*Pinus densiflora* Siebold & Zucc.), Korean white pine (*P. koraiensis* Siebold & Zucc.), Japanese larch (*Larix kaempferi* (Lamb.) Carrière), and other forests. The deciduous stands were divided into oak (*Quercus* spp.) forests and other forests. Areas with >25% but <75% of either coniferous or deciduous trees were considered as mixed forests. To exclude reduction in NDMI due to the inclusion of non-forested area, forest areas were selected based on the forest type map of the Korean Forest Service (http://www.forest.go.kr (accessed on 26 April 2021)). Forests that had been heavily thinned or artificially damaged were also excluded from the analysis. For satellite image analysis and spatial information analysis, the ENVI + IDL software (Version 5.5, Harris Geospatial, Boulder, CO, USA) and ArcGIS (Version 10.4, ESRI, Redlands, CA, USA) were used.

*2.3. Field Survey*

To validate the results of the analysis of satellite images, a field survey was conducted at Wonju on 17 June 2020. At first, information on stands that were heavily defoliated by gypsy moth was collected in the center of Wonju, where the damage caused by gypsy moth was concentrated. Stands that are heavily defoliated look red and the density of gypsy moth larvae there was extremely high. These stands were considered as defoliated stands and were delimited based on forest address and forest type map according to the tree species defoliated. In addition to the heavily defoliated stands, damage and the presence of gypsy moth in nearby stands were also observed. The degree of damage by gypsy moth in the field was not classified at the stand level since the level of defoliation was dependent on tree species [15] and the stands were heavily fragmented (Figure 1).

*2.4. Data Analysis*

To measure the host preference of gypsy moth, an $\chi^2$-test was conducted based on the area of each forest type and the area of forest damaged by gypsy moth. The degree of preference was estimated based on Manly's alpha feeding preference [20]. This was performed using the package selectapref [21] and R (http://cran.r-project.org (accessed on 26 April 2021)).

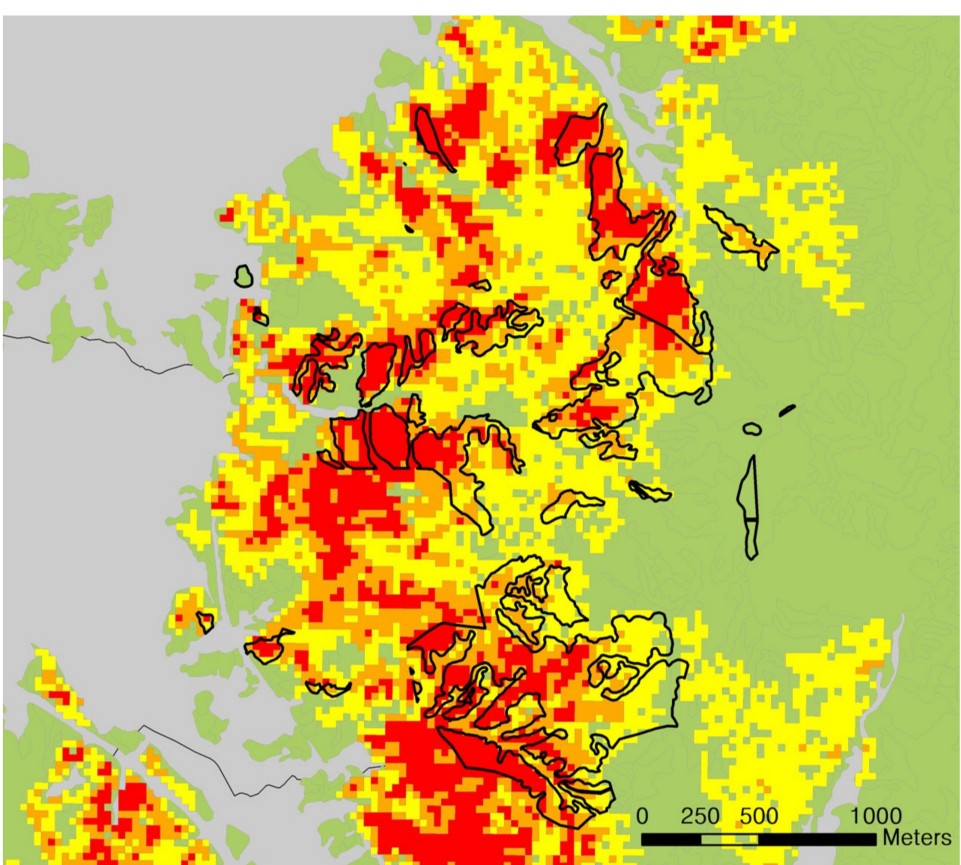

**Figure 1.** Comparison between field survey results and Landsat-based gypsy moth defoliation maps in the center of Wonju. Damage degree represents the mean reduction in Normalized Difference Moisture Index (NDMI) scores calculated using all the available observations in June from 2017 to 2019. The closed curves indicated the larch stands defoliated by gypsy moth observed through a field survey.

## 3. Results

For the total forested area (60,186 ha) in Wonju, the NDMI values of 7825 ha were reduced more than 0.05 (our criterion for at least light defoliation) when compared to the location mean NDMI values for the prior 3 years (2017 to 2019), suggesting that this area had been affected by gypsy moth defoliation (Figure 2, Table 1). The estimates of areas with moderate and heavy defoliation were 1350 and 656 ha, respectively. At least 13.0% of the forest area in Wonju was at least partially defoliated by gypsy moth.

The field survey results showed that eighteen stands were defoliated by gypsy moth and all of them were *L. kaempferi* stands (Figure 1). Among them, 93.0% of stands were judged as defoliated stands by our satellite image analysis, with severe defoliation at 37.4%, moderate defoliation at 30.6%, and light defoliation at 25.0%. In addition to the larch stands, extensive defoliation by gypsy moth was observed (Figure 2). These results suggested that the Landsat satellite image analysis using NDMI is a useful method to detect the extent and severity of defoliation caused by forest pests in Korea.

The host preferences of gypsy moth based on the satellite image analysis showed that the occurrence of gypsy moth concentrated on the specific forest type ($\chi^2$ = 759.3, df = 6, $p$ < 0.01). The preference indexes of gypsy moth ranged from 36.1 to 100 according to the forest stand type and was the highest for the larch stand. The preference for *P. koraiensis* was lowest and the preference for *Quercus* spp. was 37.6.

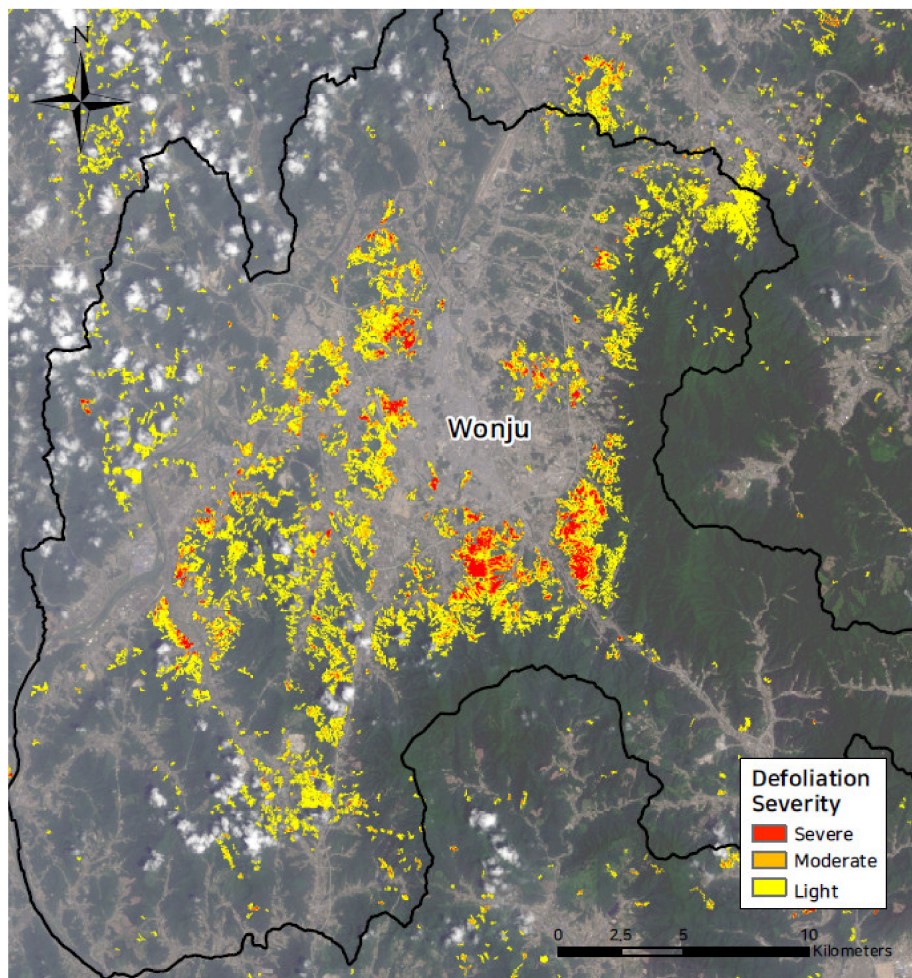

**Figure 2.** Landsat-based gypsy moth defoliation maps in Wonju and the surrounding area in 2020. Mapped results represent the mean reduction in Normalized Difference Moisture Index (NDMI) scores calculated using all the available observations in June from 2017 to 2019.

**Table 1.** Area defoliated by gypsy moth at Wonju in 2020 according to the defoliation severity and forest type. The areas were estimated based on the reduction in NDMI in 2020 compared to the prior 3 years.

| Defoliation Severity | Forest Type | | | | | | | Total |
|---|---|---|---|---|---|---|---|---|
| | *Pinus densiflora* | *Pinus koraiensis* | *Larix kaempferi* | Other Coniferous | *Quercus* Spp. | Other Deciduous | Mixed | |
| Severe | 27 (0.3%) | 14 (0.5%) | 274 (5.2%) | 24 (2.2%) | 82 (0.4%) | 178 (1.1%) | 57 (0.7%) | 656 (1.1%) |
| Moderate | 119 (1.5%) | 44 (1.6%) | 301 (5.7%) | 47 (4.4%) | 235 (1.2%) | 404 (2.5%) | 198 (2.6%) | 1350 (2.2%) |
| Light | 788 (10.1%) | 184 (6.7%) | 626 (11.8%) | 108 (10.1%) | 1457 (7.6%) | 1573 (9.7%) | 1084 (14.0%) | 5820 (9.7%) |
| None | 6843 (88.0%) | 2491 (91.1%) | 4105 (77.4%) | 894 (83.3%) | 17,511 (90.8%) | 14,139 (86.8%) | 6378 (82.6%) | 52,361 (87.0%) |
| Total | 7777 | 2734 | 5306 | 1073 | 19,285 | 16,294 | 7717 | 60,186 |

## 4. Discussion

Remote sensing, including satellite images, has been extensively used in Canada and the USA to quantify and map outbreaks of forest pests [3,5]. In Korea, remote sensing studies have been carried out mainly for oligophagous forest pests, such as pine wilt disease

and Korean Oak wilt [22,23]. Our study shows that remote sensing images, with the use of an appropriate index, are also useful for detecting the damage of polyphagous forest pests. Moreover, unexpected outbreaks of new or occasional forest pests are increasing due to climate change and invasions of exotic pests. Under these circumstances, remote sensing using satellite images is useful for detecting or monitoring outbreaks of forest pests that cause widespread defoliation. For example, in Sweden, the invasive scale insect *Physokermes inopinatus* was effectively monitored using satellite images [4].

Outbreaks of gypsy moths have been extensively studied in North America since gypsy moth is one of the most serious forest pests in the region, where it has a long history of spreading and attempted control [10]. After the first occurrence of gypsy moth in the 1860s, it dispersed eastward and south. An entomophagous fungus, *Entomophaga maimaiga*, is considered one of key natural enemies currently regulating its population [15]. In southern New England, a wide outbreak of gypsy moth was observed from 2015 to 2017, following a 30-year period without significant outbreaks. The recent outbreak was due to reduced levels of larval mortality from the fungus due to a series of dry springs from 2014 and 2016 [3]. In addition, higher winter temperatures may have increased the survival of gypsy moth eggs [24].

Interestingly, the 5.2% of the stands of Japanese larch severely damaged by gypsy moth in Korea was greater than the amount damaged for other stand types (e.g., oak at 0.4%) (Table 1). Among forests defoliated by gypsy moth, the area of Japanese larch stands was the highest (22.7% of the total of the larch stand) (Table 1). The preference index was also highest for the larch forests. The results of the field survey also showed that Japanese larch was the most heavily attacked tree species. Although deciduous stands were not identified at the species level by forest type maps or satellite images, our results showed that oak forests are less heavily defoliated than other deciduous forests, suggesting that *Quercus* spp. were not the most preferred species among deciduous species (Table 1). Previous field surveys in Wonju showed that *Euonymus alatus*, *L. kaempferi*, *P. rigida*, *Populus davidiana*, *Q. serrata*, *Prunus sargentii*, and a shrub, *Rhododendron schlippenbachii*, were severely defoliated tree species, while oaks such as *Q. acutissima*, *Q. aliena*, *Q. mongolica*, and *Q. palustris* were only moderately defoliated [15]. These results contradicted with previous studies showing that *Quercus* spp. are some of the most preferred hosts for gypsy moth [10,12]. Further studies on host preferences are necessary to clarify the host preference of Korean gypsy moth.

Our analysis showed that defoliated areas were mainly concentrated near urban areas rather than more remote mountain areas. This was also observed in Chungju, Jecheon, and Hoengseong, in the vicinity of Wonju (Figure 1). This phenomenon was similar to earlier outbreaks of gypsy moth in Korea that were observed primarily in or near urban areas [14]. It has been hypothesized that the natural enemy regulation of gypsy moth may be weaker in or near urban areas compared to more remote forests in mountains. In the case of *Lymantria mathura*, a species similar to *L. dispar*, many natural enemies such as the parasitoid *Cotesia melanosceifus* (Ratzeburg) and several entomophagous pathogens stabilized the density of *L. mathura* in Korea [25]. The activities of these natural enemies probably decreased due to dryness or disturbance in or near urban forests. The second possible explanation for the spatial pattern of gypsy moth defoliation is that urban environments offer more opportunity for increase in gypsy moth populations. The lights from urban areas attract adult moths, which may then lay their eggs locally. Moreover, the winter temperatures near urban areas can be higher than those in forested areas, and this may increase the winter survival of gypsy moth eggs.

## 5. Conclusions

Our study showed that areas defoliated by gypsy moth may be effectively detected using satellite images and that such information may help us understand the characteristics of outbreaks of gypsy moths. Gypsy moth populations in Wonju severely defoliated Japanese larch trees at higher rates than other tree species or groups. Additionally, gypsy moth

damage was concentrated in or near urban areas. Our results showed that remote sensing methods are useful for detecting or monitoring damage caused by some forest pests, such as those that occasionally reach outbreak densities or that are newly invasive species. Moreover, the defoliation maps estimated by remote sensing techniques can be basic information for outbreak risk analysis using numerical and mathematical methods [26–30].

**Author Contributions:** Conceptualization, W.-I.C., E.-S.K., and J.-H.L.; methodology, E.-S.K.; validation, W.-I.C., E.-S.K., and J.-H.L.; formal analysis, W.-I.C. and E.-S.K.; data curation, E.-S.K., S.-J.Y., and Y.-E.K.; writing—original draft preparation, W.-I.C. and E.-S.K.; writing—review and editing, W.-I.C. and E.-S.K.; visualization, S.-J.Y. and Y.-E.K. All authors have read and agreed to the published version of the manuscript.

**Funding:** This research received no external funding.

**Conflicts of Interest:** The authors declare no conflict of interest. The funders had no role in the design of the study; in the collection, analysis, or interpretation of data; in the writing of the manuscript; or in the decision to publish the results.

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
