# Peer review of "Quantification of One-Year Gypsy Moth Defoliation Extent in Wonju, Korea, Using Landsat Satellite Images"

_forests, doi:10.3390/f12050545_

Round 1
Reviewer 1 Report
I’m not a fan of the title, repetition of “of” and “in” – could be reworded. Quantification of One-Year Gypsy Moth Defoliation Extent in Wonju, Korea, Using Landsat Images.
Overall, I think it is a relevant study for publication in Forests. There are issues with the English grammar in multiple places. I’ve indicated some in my specific comments below, but the authors would benefit from ensuring they fully review the English grammar throughout the manuscript.
L35 and 36 – remove one of the “forest” – either “forest area damaged by pests” or “area damaged by forest pests”
L42, 44 – as this is an international journal, need to indicate these are in USA.
L47, 49, etc. – official common name is gypsy moth, not the gypsy moth. Remove “the” from any use of gypsy moth.
L51-52 – nonsensical – fix English grammar
L54-56 – should be “The fact that egg masses of gypsy moth were collected…”
L55, 189 – spp. Should not be italicized
L68 – remove second instance of “with”
L69 – does the highest mountain matter? Should it just be an average elevation for the area if needed?
L75 – correct to “…and its vicinity from 2017 to 2020 were obtained from…”
L82 – average? Mean? Clarify. Some indication in the results of variance between the baseline years is necessary.
L99 – Were the forest type maps raster or vector? If raster, how did resolution compare to Landsat images? If vector, was it rastorized to code forest type to individual pixels?
L113 – “…on stands that were heavily…” and “…where damage by…”
L115 – looks red in person (it’s in the field validation paragraph) or on the satellite image? Looks red because that is the symbology in the map?
L115 – Figures 1 and 2 are results, not methods. Need to be removed from this paragraph.
Figure 1 and 2 – Clarify average as mean and remove the ‘ in front of change – should change be reduction? Adds specificity to the direction of change.
L137 – properly cite both selectapref (Richardson 2020) and R.
L141 – change acreage to area
L144 – wording suggests complete removal of all leaves – maybe “…forest area in Wonju was at least partially defoliated…”
L159 – italicize Quercus
L186 – remove “tree”
L190 – not clear why Wonju is hyphenated here.
Author Response
Thank you for your encouraging comments and criticisms which will lead to greater clarity and relevance of the MS.
Reviewer # 1
I’m not a fan of the title, repetition of “of” and “in” – could be reworded. Quantification of One-Year Gypsy Moth Defoliation Extent in Wonju, Korea, Using Landsat Images.
- Response: Thank you for your advice. The title was changed according to your comments as follows: “Quantification of One-Year Gypsy Moth Defoliation Extent in Wonju, Korea, Using Landsat Satellite Images”
Overall, I think it is a relevant study for publication in Forests. There are issues with the English grammar in multiple places. I’ve indicated some in my specific comments below, but the authors would benefit from ensuring they fully review the English grammar throughout the manuscript.
-
- Response: We appreciate your advice. We have revised all of your specific comments and have re-checked the English grammar of the manuscript.
L35 and 36 – remove one of the “forest” – either “forest area damaged by pests” or “area damaged by forest pests”
-
- Response: Thank you for your advice. We have removed one of the “forests” in the sentence as follows: Quantification of area damaged by forest pests using remote sensing technologies has been extensively reviewed.
L42, 44 – as this is an international journal, need to indicate these are in USA.
- Response: Thank you for your suggestion. We have added USA after the name of state.
L47, 49, etc. – official common name is gypsy moth, not the gypsy moth. Remove “the” from any use of gypsy moth.
-
- Response: We appreciate your comment. We have removed “the” from any use of gypsy moth in manuscript.
L51-52 – nonsensical – fix English grammar
-
- Response: Thank you for your advice. We have rewritten the sentence as follow: Until 1990s, there was little information about the outbreak of gypsy moth in Korea except for a small outbreak of 1,200 deciduous trees in Seoul, 1959.
L54-56 – should be “The fact that egg masses of gypsy moth were collected…”
- Response: Thank you for your comment. We have changed the word “was” into “were” as follows: The fact that egg masses of gypsy moth were collected in mainly Quercus spp. forests in early 1990s suggested Quercus spp. potentially were preferred hosts in Korea.
L55, 189 – spp. Should not be italicized
- Response: We appreciate your advice. We have changed the word “Quercus spp.” into “Quercus spp.” throughout the manuscript.L68 – remove second instance of “with”
-
- Response: Thank you for your advice. We have removed second instance of “with” as follows: Wonju is urban areas with 350,000 residents and area of 867.30 km2.
L69 – does the highest mountain matter? Should it just be an average elevation for the area if needed?
-
- Response: Thank you for your comment. We have removed the sentence as your commented.
L75 – correct to “…and its vicinity from 2017 to 2020 were obtained from…”
- Response: We thank you for your advice. We have rewritten the sentence as follows: Wonju and its vicinity from 2017 to 2020 were obtained from USGS EarthExplorer homepage (https://earthexplorer.usgs.gov).
L82 – average? Mean? Clarify. Some indication in the results of variance between the baseline years is necessary.
- Response: Thank you for your suggestion. ‘Average’ was changed to ‘mean’. To estimate prior baseline condition, we collected images for different days and extrapolated to estimate.
L99 – Were the forest type maps raster or vector? If raster, how did resolution compare to Landsat images? If vector, was it rastorized to code forest type to individual pixels?
-
- Response: The forest type map is vector and rastorized to 30â…©30m pixels because resolution of our analysis was 30â…©30m.
L113 – “…on stands that were heavily…” and “…where damage by…”
-
- Response: Thank you for your comment. We have rewritten the sentence as follow: At first information on stands that were heavily defoliated by gypsy moth was collected in the center of Wonju where damage by gypsy moth were concentrated.
L115 – looks red in person (it’s in the field validation paragraph) or on the satellite image? Looks red because that is the symbology in the map?
- Response: We appreciate your advice. We agree that the sentence was confusing. The sentence means that it looks red by person. And We have revised the sentence clearly as follows: The heavily defoliated stand was observed red and density of gypsy moth larvae in there was extreme high.
L115 – Figures 1 and 2 are results, not methods. Need to be removed from this paragraph.
- Response: Thank you for your advice. We have removed as you commented.
Figure 1 and 2 – Clarify average as mean and remove the ‘ in front of change – should change be reduction? Adds specificity to the direction of change.
-
- Response: Thank you for your comment. We have changed “average” into “ mean” and have revised caption of Figure 1 and 2 as follows: “Figure 2. Comparison between field survey results and Landsat-based gypsy moth defoliation maps in center of Wonju. Damage degrees represent mean reduction in NDMI (Normalized Difference Moisture Index) scores calculated using all available observations in Junes from 2017 to 2019. The closed curves indicated the larch stands defoliated by gypsy moth observed through field survey.”
L137 – properly cite both selectapref (Richardson 2020) and R.
-
- “Figure 1. Landsat-based gypsy moth defoliation maps in Wonju and its vicinity areas in 2020. Mapped results represent mean reduction in NDMI (Normalized Difference Moisture Index) scores calculated using all available observations in Junes from 2017 to 2019”
- Response: We appreciate your comment. We have revised the sentence as follows: This was performed using the package selectapref [21] and R (http://cran.r-project.org).L
141 – change acreage to area
-
- Response: Thank you for your comment. We have changed “acreage” into “area” throughout the manuscript according to your comment.
L144 – wording suggests complete removal of all leaves – maybe “…forest area in Wonju was at least partially defoliated…”
- Response: We appreciate your advice. We have revised the sentence as follows: At least, 13.0% of the forest area in Wonju was at least partially defoliated by gypsy moth.
L159 – italicize Quercus
- Response: Thank you for your suggestion. We have changed the word “Quercus spp.” Into “Quercus spp.”
L186 – remove “tree”
- Response: Thank you for your comment. We have removed “tree” as you commented.
L190 – not clear why Wonju is hyphenated here.
- Response: Thank you for your advice. We have removed hyphen as you commented.

Reviewer 2 Report
In this study, the extent and severity of Asian gypsy moth (Lymantria dispar) defoliation in Wonju, Korea from May to early June in 2020 was quantified. Landsat images were collected covering Wonju and its vicinity in June, from 2017 to 2020. Forest damage was evaluated based on differences between the NDMI (Normalized Difference Moisture Index) from images acquired in 8 June 2020, and the prior average NDMI estimated from images June from 2017 to 2019.
The paper’s subject is very practical and interesting. The research procedure has been logically carried out. Therefore, I highly recommend this paper for publication in this journal but before that, I have some few comments on the text:
Comments:
1)The figure captions (specially the font) are not in correct format. Please check the journal format for figure caption.
2)What software has been used for image analysis? It would be nice if it is mentioned in the text.
3) In line 74-76, the authors mentioned that “…serial Landsat images (Path/Row 115/34) covering Wonju and its vicinity were collected from 2017 to 2020 were obtained from USGS EarthExplorer homepage(https://earthexplorer.usgs.gov).” My question is that how many images/dataset in the time period of 2017-2020 have been used in this study?
4) In recent years, numerical and mathematic methods have been widely used as a powerful tool for analysis of data in field of forestry and biology which. I recommend the authors to add some more new references in this field in the paper. It would be certainly interesting for the readers of the journal. Some suitable references that are highly recommended are listed in the following:
[1]Sheng He, Lanying Lin, Zaixing Wu, Zhangmin Chen. Application of Finite Element Analysis in Properties Test of Finger-jointed Lumber[J]. Journal of Bioresources and Bioproducts, 2020, 5(2): 124-133
[2] Dargahi, M., Newson, T. and R Moore, J., 2020. A Numerical Approach to Estimate Natural Frequency of Trees with Variable Properties. Forests, 11(9), p.915.
[3]Kayet N, Chakrabarty A, Pathak K, et al. Comparative analysis of multi-criteria probabilistic FR and AHP models for forest fire risk (FFR) mapping in Melghat Tiger Reserve (MTR) forest[J]. Journal of Forestry Research, 2020, 31(2): 565-579.
[4]S.E .Ibitoye, I.K. Adegun, P.O. Omoniyi, T.S. Ogedengbe, O.O. Alabi. Numerical investigation of thermo-physical properties of non-newtonian fliud in a modelled intestine[J]. Journal of Bioresources and Bioproducts, 2020, 5(3): 211-221.
Author Response
Thank you for your encouraging comments and criticisms which will lead to greater clarity and relevance of the MS.
In this study, the extent and severity of Asian gypsy moth (Lymantria dispar) defoliation in Wonju, Korea from May to early June in 2020 was quantified. Landsat images were collected covering Wonju and its vicinity in June, from 2017 to 2020. Forest damage was evaluated based on differences between the NDMI (Normalized Difference Moisture Index) from images acquired in 8 June 2020, and the prior average NDMI estimated from images June from 2017 to 2019.
The paper’s subject is very practical and interesting. The research procedure has been logically carried out. Therefore, I highly recommend this paper for publication in this journal but before that, I have some few comments on the text:
Comments:
1)The figure captions (specially the font) are not in correct format. Please check the journal format for figure caption.
- Response: Thank you for your comment. We have re-check the journal format for figure caption.
2)What software has been used for image analysis? It would be nice if it is mentioned in the text.
- Response: Thank you for your advice. We have added the software has been used for image analysis.
3) In line 74-76, the authors mentioned that “…serial Landsat images (Path/Row 115/34) covering Wonju and its vicinity were collected from 2017 to 2020 were obtained from USGS EarthExplorer homepage(https://earthexplorer.usgs.gov).” My question is that how many images/dataset in the time period of 2017-2020 have been used in this study?
- Response: We appreciate your advice. Basically, one scene of Landsat covered Wonji city and its vicinity. However, 20 images per year were collected to estimate baseline information for each year. Therefore about 60 images were collected to estimate total baseline for our analysis. Please see below diagram that was published by Kim et al. (2019).
4) In recent years, numerical and mathematic methods have been widely used as a powerful tool for analysis of data in field of forestry and biology which. I recommend the authors to add some more new references in this field in the paper. It would be certainly interesting for the readers of the journal. Some suitable references that are highly recommended are listed in the following:
- Response: Thank you for your comment. We have added new reference in this field in the paper as you commented. We have added two of them in last sentence. [3]Kayet N, Chakrabarty A, Pathak K, et al. Comparative analysis of multi-criteria probabilistic FR and AHP models for forest fire risk (FFR) mapping in Melghat Tiger Reserve (MTR) forest[J]. Journal of Forestry Research, 2020, 31(2): 565-579.
- [2] Dargahi, M., Newson, T. and R Moore, J., 2020. A Numerical Approach to Estimate Natural Frequency of Trees with Variable Properties. Forests, 11(9), p.915.

Round 2
Reviewer 2 Report
Most of comments have been addressed correctly, but some of them have been responded incompletely that should be replied before publication:
Comments:
- A) The response of authors to this question: “In line 74-76, the authors mentioned that “…serial Landsat images (Path/Row 115/34) covering Wonju and its vicinity were collected from 2017 to 2020 were obtained from USGS EarthExplorer homepage(https://earthexplorer.usgs.gov).” My question is that how many images/dataset in the time period of 2017-2020 have been used in this study?” is convincing, but I don’t see the response in the text. It would be important for the readers to know about that. Please add it in the text and highlight it.
- B) In response to my comment; “In recent years, numerical and mathematic methods have been widely used as a powerful tool for analysis of data in field of forestry and biology which. I recommend the authors to add some more new references in this field in the paper. It would be certainly interesting for the readers of the journal. Some suitable references that are highly recommended are listed in the following:”, authors only added the first 2 references. I highly recommend the authors to add all the recommended citations, references 3 and 4 (shown following) would be indeed interesting to journal’s reader.
[3]Kayet N, Chakrabarty A, Pathak K, et al. Comparative analysis of multi-criteria probabilistic FR and AHP models for forest fire risk (FFR) mapping in Melghat Tiger Reserve (MTR) forest[J]. Journal of Forestry Research, 2020, 31(2): 565-579.
[4]S.E .Ibitoye, I.K. Adegun, P.O. Omoniyi, T.S. Ogedengbe, O.O. Alabi. Numerical investigation of thermo-physical properties of non-newtonian fliud in a modelled intestine[J]. Journal of Bioresources and Bioproducts, 2020, 5(3): 211-221.
Author Response
Thank you for giving one more chance to correct MS.
Comments:
1) The response of authors to this question: “In line 74-76, the authors mentioned that “…serial Landsat images (Path/Row 115/34) covering Wonju and its vicinity were collected from 2017 to 2020 were obtained from USGS EarthExplorer homepage(https://earthexplorer.usgs.gov).” My question is that how many images/dataset in the time period of 2017-2020 have been used in this study?” is convincing, but I don’t see the response in the text. It would be important for the readers to know about that. Please add it in the text and highlight it.
- Response: We appreciate your advice. Basically, one scene of Landsat covered Wonji city and its vicinity. To estimate baseline information for each year, 20 images per year were collected. Therefore, 60 images were collected to estimate three prior years baseline information. We added this explanation in the manuscript and highlight it.
2) In response to my comment; “In recent years, numerical and mathematic methods have been widely used as a powerful tool for analysis of data in field of forestry and biology which. I recommend the authors to add some more new references in this field in the paper. It would be certainly interesting for the readers of the journal. Some suitable references that are highly recommended are listed in the following:”, authors only added the first 2 references. I highly recommend the authors to add all the recommended citations, references 3 and 4 (shown following) would be indeed interesting to journal’s reader.
- Response: Thank you for your comment. We have added all of references that you recommended.
- Dargahi, M.; Newson, T.; Moore, J.R. A Numerical Approach to Estimate Natural Frequency of Trees with Variable Properties. Forests, 2020, 11(9), 915.
- Kayet, N.; Chakrabarty, A.; Pathak, K.; Sahoo, S.; Dutta, T.; Hatai, B.K. Comparative analysis of multi-criteria probabilistic FR and AHP models for forest fire risk (FFR) mapping in Melghat Tiger Reserve (MTR) forest. Journal of Forestry Research, 2020, 31(2), 565-579.
- He, S.; Lin, L.; Wu, Z.; Chen, Z. Application of Finite Element Analysis in Properties Test of Finger-jointed Lumber. Journal of Bioresources and Bioproducts, 2020, 5(2), 124-133.
- Ibitoye, E.; Adegun, I.K.; Omoniyi, P.O.; Ogedengbe, T.S.; Alabi, O.O. Numerical investigation of thermo-physical properties of non-newtonian fliud in a modelled intestine. Journal of Bioresources and Bioproducts, 2020, 5(3), 211-221.
